# FBXO10 Drives Hepatocellular Carcinoma Proliferation via K63-Linked Ubiquitination and Stabilization of FRMPD1

**DOI:** 10.3390/cimb47060391

**Published:** 2025-05-24

**Authors:** Wuguang Liu, Bin Xu, Kashif Kifayat, Yuhong Xie, Xiaolong Liu, Chengyong Dong, Liming Wang

**Affiliations:** 1Division of Hepatobiliary and Pancreatic Surgery, Department of General Surgery, The Second Hospital of Dalian Medical University, Dalian 116027, China; 2Institute of Cancer Stem Cell, Dalian Medical University, Dalian 116044, China

**Keywords:** hepatocarcinoma, FBXO10, FRMPD1, ubiquitination, cell proliferation

## Abstract

Aberrant ubiquitination drives hepatocellular carcinoma (HCC) progression, yet the role of FBXO10—a key F-box E3 ubiquitin ligase component—remains uncharacterized. Through bioinformatics analyses and functional validation, we establish FBXO10 as a critical oncogenic driver in HCC. Transcriptomic data from public databases (TIMER, UALCAN, GEO) revealed significant FBXO10 upregulation in HCC tissues, with elevated expression predicting advanced tumor stage, metastasis, and reduced survival. Functionally, FBXO10 silencing suppressed HCC cell proliferation while its overexpression promoted tumor growth. Mechanistic studies revealed that FBXO10 directly interacts with FRMPD1 to mediate its K63-linked polyubiquitination and stabilization, independent of transcriptional regulation. FRMPD1 restoration rescued FBXO10-mediated proliferation, confirming its role as the key downstream effector. Clinically, FBXO10 expression correlated with TP53 mutations and adverse clinicopathological features. Our findings reveal a novel FBXO10–FRMPD1 axis promoting hepatocarcinogenesis through post-translational stabilization, positioning FBXO10 as both a prognostic biomarker and therapeutic target in HCC.

## 1. Introduction

Hepatocellular carcinoma (HCC) is classified as the sixth most common malignant neoplasm globally and serves as the third primary contributor to cancer-associated deaths worldwide [1]. Clinicaloutcomes continue to be unfavorable, primarily due to difficulties in early detection and the limited effectiveness of current treatment regimens [2]. Although the introduction of molecular-targeted agents and immune checkpoint inhibitors has enhanced the survival rates for certain HCC patients, a substantial proportion of patients eventually acquire therapeutic resistance, which may be associated with the intricate nature and cellular diversity within the tumor microenvironment [3,4,5,6,7,8]. Therefore, delineating the pathogenic molecular pathways driving hepatic carcinogenesis and identifying novel treatment approaches focused on modulating the hepatic neoplastic niche become critical. Such strategies could enable the development of innovative combination therapies to effectively inhibit tumor progression or reverse immunotherapy resistance in refractory cases [9,10,11].

F-box proteins constitute a conserved protein family responsible for mediating substrate recognition in the SCF (Skp1-Cullin-F-box) E3 ubiquitin ligase complex. These proteins are structurally characterized by an F-box motif that facilitates binding with Skp1, complemented by diverse substrate-binding domains that determine target specificity [12]. Extensive studies in cellular systems, murine models, and human malignancies have revealed the critical physiological roles of F-box proteins. Notably, multiple members function as direct oncoproteins/tumor suppressors or exert indirect regulatory effects on carcinogenesis. Their tumor-modulating activities primarily stem from the ubiquitination-mediated regulation of cancer-associated substrates, particularly those governing fundamental oncogenic processes such as cell cycle control; DNA repair mechanisms; EMT regulation; and key signaling cascades such as the PI3K/AKT, BMP, p53, NRF2, AMPK/mTOR, NF-κB, and Hippo pathways. Through these molecular interactions, F-box proteins critically influence tumor development, expansion, metastatic dissemination, and invasive potential [13].

FBXO10 represents a critical component of F-box proteins that exhibit ubiquitin E3 ligase functionality, as documented in previous studies [12,14]. Research has demonstrated its capacity to interact with the anti-apoptotic protein BCL-2 in human lymphoma systems, subsequently promoting its proteasomal degradation [14]. Emerging evidence suggests that FBXO10 may serve as a tumor suppressor in lymphoid malignancies [15]. Furthermore, FBXO10 has been shown to bind to the human germinal center-associated lymphoma (HGAL) protein, mediating its ubiquitination and subsequent degradation processes implicated in lymphoma pathogenesis and immune regulation [16]. Despite these findings, the functional significance of FBXO10 in HCC progression remains poorly characterized. This investigation aims to systematically elucidate the biological roles and underlying molecular mechanisms through which FBXO10 regulates HCC pathogenesis, ultimately seeking to identify novel therapeutic strategies for HCC management.

## 2. Materials and Methods

### 2.1. TIMER Database Analysis

The Tumor Immune Estimation Resource (TIMER) was used as the primary platform to assess transcriptional variations in FBXO10 expression between tumor and matched normal tissues across 33 cancer types [17]. This analysis provided a pan-cancer overview of FBXO10 dysregulation. The “Correlation” module within TIMER facilitated initial screening of associations between FBXO10 levels and overlapping genes (identified through intersection analysis), with statistical significance set at *p* < 0.05. Data from TIMER were used independently to prioritize HCC for further investigation.

### 2.2. UALCAN Database Analysis

To validate and extend TIMER findings in HCC, UALCAN [18] was employed to analyze TCGA-derived gene expression and clinical datasets. This independent platform allowed detailed exploration of FBXO10 mRNA/protein expression patterns in primary hepatocellular carcinoma tissues and their relationships with clinicopathological parameters.

### 2.3. GEO Database Validation

To ensure robustness, FBXO10 expression patterns identified in TCGA (via TIMER/UALCAN) were independently validated using the GSE76427 cohort from the Gene Expression Omnibus (GEO) repository [19]. No integration of raw data occurred; GEO datasets were analyzed separately using R software (v4.0.3), with Wilcoxon tests for group comparisons (*p* < 0.05).

### 2.4. Human Protein Atlas (HPA)

The FBXO10 mRNA/protein expression patterns and their clinical correlations in HCC were examined using the HPA database (https://www.proteinatlas.org/ accessed on 4 November 2024) [20].

### 2.5. LinkedOmics Database Exploration

The LinkedOmics platform integrates multi-omics data from 11,158 patients spanning 32 cancers [21]. Through its “LinkFinder” module, we identified FBXO10-associated differentially expressed genes (DEGs) in HCC. Pearson correlation analyses were conducted, with the results visualized through volcano plots and heatmaps.

### 2.6. cBioPortal Genomic Profiling

The cBioPortal resource (https://www.cbioportal.org/ accessed on 4 November 2024), designed for cancer genomics exploration [22], enabled an analysis of the FBXO10 genomic alterations in HCC using the CLCA (Nature 2024) dataset.

### 2.7. Protein–Protein Interaction (PPI) Analysis

PPI networks for FBXO10 were constructed via the STRING database using moderate confidence thresholds (score > 0.4) and limiting interactors to 10 nodes. A Venn diagram analysis showed that the FBXO10-associated genes intersected with its interaction partners.

### 2.8. Cell Lines and Culture Conditions

The HCC cell lines HCC-LM3, MHCC-97H, HepG2, and Hep3B were obtained from the Type Culture Collection Committee of the Chinese Academy of Sciences (Shanghai, China). The HEK-293T cell line was acquired from the Type Culture Collection Committee of the Chinese Academy of Sciences (Shanghai, China). The HCC-LM3, MHCC-97H, HepG2, Hep3B and HEK-293T cell lines were profiled and authenticated using the short tandem repeat identification criteria developed by the International Cell Line Authentication Committee [23]. The HCC-LM3, HepG2, Hep3B and HEK-293T cells were cultured in Dulbecco’s Modified Eagle Medium (DMEM; Gibco, New York, NY, USA) supplemented with 10% fetal bovine serum (FBS; Gibco, NY, USA) and 1% penicillin-streptomycin (Gibco, NY, USA). The MHCC-97H cells were grown in RPMI 1640 medium (Gibco, NY, USA) containing 10% FBS and 1% penicillin-streptomycin (Gibco, NY, USA). All cell lines were maintained at 37 °C in a humidified 5% CO_2_ incubator.

### 2.9. Antibodies and Chemicals

MG132 (#S2619) and cycloheximide (CHX, #2112S) was purchased from Selleck Chemicals (Houston, TX, USA). The primary antibodies used were FBXO10 (1:1000, Invitrogen, CA, US, #PA5-113506), GAPDH (1:1000, Proteintech, Wuhan, China, #10494-1-AP), Myc (1:1000, Proteintech, WH, CN, #60003-2-Ig), Flag (1:1000, Sigma, St. Louis, MO, USA, #F7425), HA (1:1000, Santa Cruz Biotechnology, Santa Cruz, CA, USA, #sc-7392), and FRMPD1 (1:1000, Sigma, STL, US, #SAB1407148). The HRP-conjugated anti-mouse/anti-rabbit secondary antibodies (1:5000; Cell Signaling Technology, Danvers, MA, USA, #7076 and #7074).

### 2.10. Plasmids

Gene-specific shRNAs targeting FBXO10 (shFBXO10-1/2) and FRMPD1 (shFRMPD1-1/2), along with a non-targeting control (pGIPZ-shControl), were synthesized by GenePharma (Suzhou, China). Full-length human FBXO10 and FRMPD1 cDNA sequences were cloned into pDONOR vectors using Gateway Technology (Thermo Fisher Scientific, Waltham, MA, USA) to generate entry clones. These were subsequently recombined into FLAG- or Myc-tagged Gateway destination vectors for protein expression.

### 2.11. Method Section Addition: Construction of Stable Cell Lines

To establish stable FBXO10-knockdown or FRMPD1-overexpressing HCC cell lines, lentiviral transduction was performed. For FBXO10 knockdown, short hairpin RNAs (shRNAs) targeting human FBXO10 (shFBXO10-1/2) and a non-targeting control (shControl) were cloned into the pLKO.1-puro vector (Addgene). For FRMPD1 overexpression, full-length FRMPD1 cDNA was subcloned into the pLVX-IRES-puro vector (Clontech, San Jose, CA, USA). Lentiviral particles were produced by co-transfecting HEK-293T cells with the respective plasmids and packaging vectors (psPAX2 and pMD2.G) using Lipofectamine 3000 (Thermo Fisher Scientific). Viral supernatants were collected at 48 and 72 h post-transfection, filtered, and used to infect HCC cells in the presence of 8 μg/mL polybrene (Sigma-Aldrich, St. Louis, MO, USA). Stable cell pools were selected with 2 μg/mL puromycin (InvivoGen, Toulouse, France) for 7 days. Knockdown and overexpression efficiencies were validated via quantitative Western blotting. Cells were maintained in complete medium supplemented with 1 μg/mL puromycin to ensure stable expression.

### 2.12. Western Blot Analysis

Total protein lysates were obtained from transfected cells through homogenization in a RIPA buffer, with protein quantification performed using the Thermo Scientific™ BCA Assay Kit (Thermo Fisher Scientific). The protein samples underwent thermal denaturation (95 °C, 5 min) in a boiling water bath before electrophoretic separation on 10% SDS-polyacrylamide gels. Subsequent electroblotting transferred the proteins onto PVDF membranes (0.45 μm pore size). The membranes were first blocked with 5% non-fat dry milk in TBST for 60 min at ambient temperature and then probed with the primary antibodies during overnight incubation at 4 °C. Following three TBST washes (10 min each), the membranes were exposed to HRP-conjugated secondary antibodies for 1 h at room temperature. After additional TBST washes, their chemiluminescence was detected using a Biyuntian ECL substrate (Shanghai Biyuntian Bio-Technique, Shangai, China), with the signal acquired from a LI-COR Odyssey imaging system.

### 2.13. Crystal Violet Proliferation Assay

The cells were seeded at 3000 cells/well in 6-well plates. At each time point (3, 5, 7, 9, and 11 days post-seeding), cultures were fixed with 4% paraformaldehyde in PBS for 15 min at room temperature, washed twice with PBS, and subjected to crystal violet staining. The stained cells were dissolved in acetic acid, and absorbance at 590 nm was measured using a Biotek Synergy H1 Hybrid Reader (Agilent Technologies, Santa Clara, CA, USA). Experiments included triplicate biological replicates.

### 2.14. Co-Immunoprecipitation (Co-IP) and Affinity Purification

The cells were lysed in an NETN lysis buffer. An antibody against FBXO10 was used for immunoprecipitation. For the Flag bead pulldown experiment, the HEK-293T cells were transfected with Flag-tagged protein and lysed in the NETN buffer for 20 min at 4 °C. The crude lysates were subjected to centrifugation at 12,000× *g* for 15 min at 4 °C, and the supernatants were incubated with an Anti-Flag Affinity Gel (Bimake, B23102, Houston, TX, USA) for 4 h. Then, the agaroses were washed three times with the NETN buffer. Finally, the proteins were eluted by boiling in a 1× SDS running buffer and subjected to SDS/PAGE for immunoblotting.

### 2.15. Ubiquitination Analysis

The Cells were lysed in Laemmli buffer (2% SDS, 10% glycerol, 60 mM Tris-HCl pH 6.8, 0.02% bromophenol blue), heat-denatured at 95 °C, and diluted with NP-40 lysis buffer (1% NP-40, 150 mM NaCl, 50 mM Tris-HCl pH 7.4) to reduce SDS concentration. The lysates were immunoprecipitated with an HA-ubiquitin antibody and analyzed via immunoblotting using HA, Myc, or Flag antibodies.

### 2.16. Reverse Transcription-Quantitative PCR (RT-qPCR)

Total RNA was isolated from cellular and tissue samples utilizing a TRIpure Reagent (BioTeke Corporation). Then, a subsequent cDNA synthesis was conducted with BeyoRT II M-MLV reverse transcriptase (Beyotime Institute of Biotechnology, Shanghai, China) or the miRNA First Strand cDNA Synthesis Kit (Tailing Reaction, cat# B532451; Sangon Biotech) following the manufacturer’s instructions. A quantitative PCR analysis was carried out on an Exicycler™ 96 platform (Bioneer Corporation, Daejeon, Republic of Korea) employing SYBR Green detection (Solarbio Science, Beijing, China) combined with 2×Taq PCR MasterMix (Solarbio Science, Beijing, CN). The thermal cycling protocol consisted of initial DNA denaturation at 95 °C for 5 min, followed by 40 amplification cycles: 95 °C for 10 s (denaturation), 60 °C for 10 s (annealing), and 72 °C for 15 s (extension). The mRNA expression data were standardized against a GAPDH endogenous control. The relative quantification of target transcripts in the cultured cells was determined through a 2−ΔΔCq analysis [24]. The forward and reverse primers for FRMPD1 were 5′-TGCGACACACAGTAAAGATAGAC-3′ and 5′-GAGAATATCGACTGCTCGTTCC-3′, respectively. The forward and reverse primers for GAPDH were 5′-GGAGCGAGATCCCTCCAAAAT-3′ and 5′-GGCTGTTGTCATACTTCTCATGG-3′, respectively.

### 2.17. CCK-8 Cell Viability Assay

Cell proliferation and viability were assessed using a CCK-8 Assay Kit (Nanjing KeyGen Biotech Co., Ltd., Nanjing, China), according to the manufacturer’s instructions. The cells (1500/well) were plated in 96-well plates. At the indicated time points, the CCK-8 reagent was added, and absorbance at 450 nm was recorded using a microplate reader (Biotek; Agilent Technologies, Inc., Santa Clara, CA, USA).

### 2.18. Illustrations

The illustrations in the schematic diagrams were provided by Fig draw 2.0.

### 2.19. Statistical Analysis

The data between two groups were compared using Student’s *t*-test. Multiple group comparisons were analyzed via one-way ANOVA with Bonferroni post hoc testing. The associations between variables were examined using Pearson’s correlation coefficient for parametric data and Spearman’s rank correlation coefficient for nonparametric data. A survival analysis employing Kaplan–Meier curves with log-rank testing was also performed. All analyses were conducted using SPSS 18.0, with *p* < 0.05 considered statistically significant.

## 3. Results

### 3.1. FBXO10 Exhibits Marked Upregulation and Is Strongly Correlated with Unfavorable Clinical Outcomes in Hepatocellular Carcinoma

To assess the FBXO10 expression patterns across human malignancies, we analyzed tumor and normal tissue data from the TIMER database. Significant FBXO10 overexpression was observed in multiple cancers compared with that in corresponding normal tissues, including bladder urothelial carcinoma (BLCA), breast invasive carcinoma (BRCA), cervical squamous cell carcinoma/endocervical adenocarcinoma (CESC), cholangiocarcinoma (CHOL), colon adenocarcinoma (COAD), esophageal carcinoma (ESCA), head–neck squamous cell carcinoma (HNSC), HCC (LIHC), lung adenocarcinoma (LUAD), lung squamous cell carcinoma (LUSC), rectal adenocarcinoma (READ), stomach adenocarcinoma (STAD), and uterine corpus endometrial carcinoma (UCEC). Conversely, FBXO10 showed significant downregulation in kidney renal clear cell carcinoma (KIRC) and thyroid carcinoma (THCA) tissues (Figure 1A). For HCC-specific validation, an integrated analysis of the UALCAN and GEO datasets revealed substantially elevated FBXO10 mRNA and protein levels in the tumor versus normal liver tissues (Figure 1B,C). Supporting these findings, immunohistochemical data from the HPA database confirmed enhanced FBXO10 protein expression in the HCC specimens (Figure 1D). A survival analysis using TCGA data through UALCAN demonstrated that high FBXO10 expression correlated with reduced disease-free survival and overall survival compared with those in the low-expression cohorts (Figure 1E). These collective findings position FBXO10 as a promising prognostic biomarker candidate in HCC pathogenesis and clinical management.

### 3.2. Association of FBXO10 Expression with Clinicopathological Features in HCC Patients

To investigate the association between FBXO10 expression and clinicopathological characteristics in HCC patients, analyses were performed using data from the TCGA database. The UALCAN platform was employed to evaluate the correlations between FBXO10 mRNA expression levels and various clinical parameters. Elevated FBXO10 expression levels were notably observed across various clinicopathological parameters, including tumor stage, ethnicity, sex, age, body weight, histological grade, tumor histology, lymph node metastasis status, and TP53 mutation profile (Figure 2). These findings demonstrate that upregulated FBXO10 expression exhibits significant correlations with multiple clinicopathological features in HCC patients.

### 3.3. FBXO10 Exhibits Oncogenic Properties During Hepatocellular Carcinoma Proliferation

Functional analyses employing both overexpression and knockdown strategies were performed across various HCC cell models (Figure 3A,D). The forced FBXO10 expression in the HepG2 and Hep3B cell lines markedly increased cellular viability (Figure 3B,C). In contrast, RNA interference-mediated FBXO10 silencing using two independent shRNAs substantially reduced the proliferative capacity in the HCC-LM3 and MHCC-97H cells (Figure 3E,F). These complementary genetic interventions demonstrate that FBXO10 functions as a growth-promoting oncogene in hepatocellular carcinoma progression.

### 3.4. PPI Network Analysis of FBXO10-Related Genes

To investigate the biological role of FBXO10 in HCC, the co-expression network of FBXO10 was analyzed using the “Link Finder v1.0” tool within the LinkedOmics platform. The analysis identified 11,616 genes exhibiting positive correlations with FBXO10, compared with 8306 genes demonstrating negative correlations (Figure 4A). Heatmaps were generated to visualize the top 50 positively and negatively associated genes (Figure 4B,C). To elucidate the potential mechanisms of FBXO10 in HCC pathogenesis, a protein–protein interaction (PPI) network analysis was conducted via the STRING database, revealing the top 10 functional interactors of FBXO10 (Figure 4D). Cross-referencing the co-expressed genes with direct interaction partners identified nine overlapping candidates (Figure 4E). Notably, FBXO10 displayed a significant positive correlation with FRMPD1 expression (*p* <  0.001) (Figure 4F). Additionally, the associations between these nine coregulated genes and malignancy-associated gene signatures across multiple cancer types were systematically evaluated (Figure 4G). These findings collectively highlight a robust regulatory relationship between FBXO10 and FRMPD1 in HCC progression.

### 3.5. FBXO10 Facilitates K63-Linked Polyubiquitination of FRMPD1 to Stabilize Its Protein Levels

To confirm the direct molecular interaction, we ectopically expressed Flag-FBXO10 in the HEK-293T cells for reciprocal co-IP assays, which consistently demonstrated FBXO10–FRMPD1 binding (Figure 5A). Their endogenous interaction was additionally confirmed in the HepG2 cells using a co-immunoprecipitation analysis (Figure 5B). Given FBXO10′s established role as an E3 ubiquitin ligase that facilitates tumorigenesis through substrate-specific ubiquitination [25], we examined its regulatory effect on FRMPD1 ubiquitination. Triple transfection experiments with Flag-FRMPD1, Myc-FBXO10, and HA-ubiquitin plasmids showed that FBXO10 overexpression markedly increased their FRMPD1 ubiquitination levels (Figure 5C). As K48-linked polyubiquitination is primarily associated with proteasomal degradation while K63-linked chains typically regulate protein stabilization or functional activation [26,27,28], we investigated the ubiquitin chain topology next. With the use of linkage-specific ubiquitin mutants in co-transfection systems, an anti-Flag immunoprecipitation analysis revealed that FBXO10 specifically promotes the K63-linked polyubiquitination of FRMPD1 (Figure 5D), indicating a potential stabilization mechanism. Consistently, FBXO10 expression substantially increased the abundance of FRMPD1 proteins (Figure 5E) without affecting its mRNA levels (Figure 5F,G). The protein stabilization effect was further confirmed through cycloheximide chase experiments showing that FBXO10 extended the protein half-life of FRMPD1 (Figure 5H,I). These collective findings demonstrate that FBXO10 enhances FRMPD1 stability via K63-linked polyubiquitination.

### 3.6. FBXO10 Enhances Hepatocellular Carcinoma Growth by Modulating FRMPD1

An evaluation via CCK-8 assays revealed that FRMPD1 knockdown significantly impaired the proliferation capabilities in the HCC-LM3 and MHCC-97H cell models (Figure 6A–C). To elucidate FRMPD1′s involvement in FBXO10-mediated proliferative regulation, both cell lines were co-transfected with FBXO10 overexpression plasmids and FRMPD1-targeting shRNA constructs (Figure 6D). Strikingly, Silencing of FRMPD1 not only markedly suppressed baseline HCC cell proliferation but also abolished the pro-proliferative effects induced by FBXO10 overexpression, as demonstrated by CCK-8 assays (Figure 6E,F). These comprehensive results identify FRMPD1 as an essential downstream effector through which FBXO10 executes its pro-proliferative function in HCC cells.

## 4. Discussion

As a central executor of protein homeostasis, the ubiquitin-proteasome system (UPS) coordinates precise regulation of tumor biology through sequential E1-E2-E3 enzymatic cascades and counterbalancing deubiquitinases (DUBs) [29,30]. Dysregulation of ubiquitination machinery contributes to hepatic carcinogenesis by inducing pathological protein aggregation and disrupting critical signaling networks [31,32,33]. Notably, FBXO10 emerges as a prominently upregulated E3 ligase in HCC, driving oncogenesis through K63-linked ubiquitination-mediated stabilization of FRMPD1. Recent advances further position ubiquitination/deubiquitination cycles as key regulators of cancer hallmarks such as senescence bypass, microbiota interactions, and cellular plasticity, establishing UPS modulation as a strategic frontier in HCC therapeutics [34,35].

The progression of HCC involves regulatory dynamics of ubiquitination-mediated post-translational modifications [36]. Diverging from canonical F-box protein functions in proteasomal substrate degradation, FBXO10 exhibits a non-degradative oncogenic mechanism in HCC by stabilizing FRMPD1 through K63-linked polyubiquitination—a functional dichotomy contrasting with its tumor-suppressive role in lymphoid malignancies [14]. This stabilization elevates FRMPD1 protein levels through post-translational regulation rather than transcriptional control, exemplifying ubiquitination’s emerging role in non-proteolytic signaling modulation. While F-box family members like FBXL16 predominantly regulate proteasome-dependent processes in cancers, the FBXO10–FRMPD1 axis defines a unique oncogenic pathway in hepatic malignancies [37]. Intriguingly, FRMPD1 demonstrates tissue-specific functional duality: it suppresses lung cancer via Hippo pathway activation yet drives HCC proliferation, highlighting the complexity of cancer signaling plasticity and metabolic reprogramming across tissue microenvironments [38].

FRMPD1 epitomizes context-dependent oncoprotein regulation, exhibiting tumor-suppressive Hippo pathway activation in lung cancer versus pro-proliferative activity in HCC [38]. Originally characterized as an AGS3-binding partner through its C-terminal TPR interaction domain, FRMPD1’s multifaceted roles in carcinogenesis remain incompletely mapped [39]. In HCC, FRMPD1 knockdown significantly impairs proliferation, contrasting starkly with its growth-inhibitory function in pulmonary malignancies. This functional reversal underscores how tissue-specific signaling architectures and microenvironmental cues reshape ubiquitination-dependent oncogenic programs. The mechanistic basis for FRMPD1’s paradoxical activities may involve differential binding partners, post-translational modifications, or crosstalk with organ-specific metabolic pathways. Elucidating FRMPD1’s interactome, downstream effectors, and cross-modulation with immune checkpoints could unravel its tissue-selective functions, providing a framework for developing context-aware therapeutic strategies in precision oncology.

While this study identifies the FBXO10–FRMPD1 axis as a critical driver of HCC proliferation via K63-linked ubiquitination, several limitations warrant further investigation. The reliance on in vitro models and bioinformatics analyses necessitates validation in in vivo HCC models to confirm the oncogenic role of FBXO10 in tumorigenesis and metastasis. The downstream signaling mechanisms through which stabilized FRMPD1 promotes proliferation remain uncharacterized, particularly its potential crosstalk with oncogenic pathways or metabolic regulators. Additionally, clinical correlations derived from public databases require prospective validation in larger, multi-ethnic cohorts to assess the prognostic utility of FBXO10/FRMPD1 expression. Future studies should evaluate the therapeutic potential of targeting the FBXO10–FRMPD1 interaction using small-molecule inhibitors or ubiquitination-modulating agents. Elucidating the interplay between this axis and the tumor immune microenvironment could further inform combinatorial immunotherapeutic strategies for HCC.

## 5. Conclusions

In summary, this study identifies FBXO10 as a pivotal regulator governing FRMPD1-dependent hepatocellular carcinogenesis through the following key findings: Elevated FBXO10 expression shows a significant clinical association with poor prognosis in HCC patients; functional validation establishes its cancer-promoting function in liver malignancies, and mechanistically, FBXO10 directly triggers the K63-linked polyubiquitination of FRMPD1 to accelerate its proteasome-dependent degradation, while the FBXO10–FRMPD1 regulatory axis critically drives hepatoma cell proliferation. This collective evidence delineates a novel pathogenic mechanism underlying HCC progression.

## Figures and Tables

**Figure 1 cimb-47-00391-f001:**
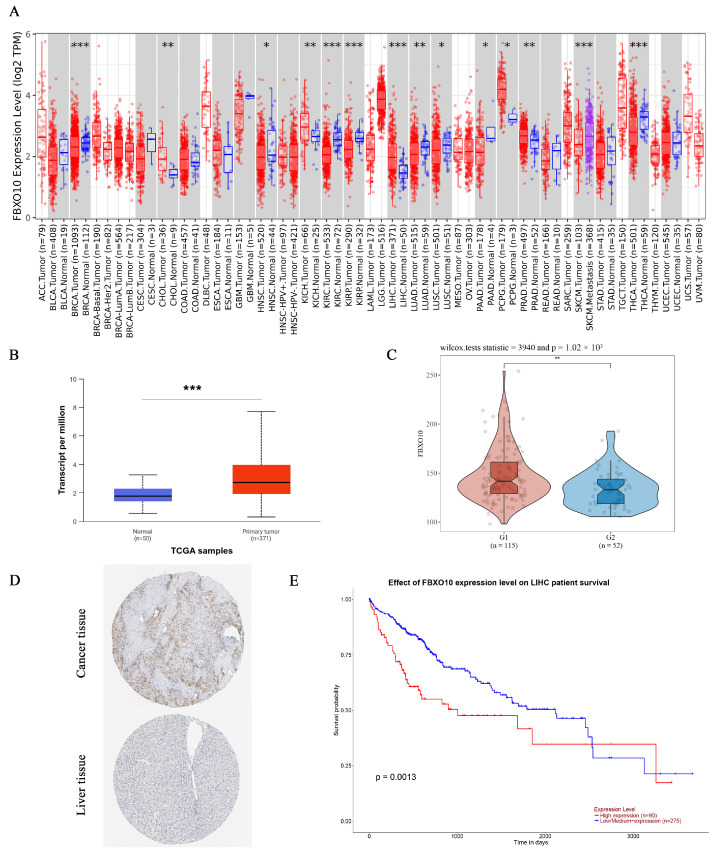
FBXO10 exhibits elevated expression and correlates with unfavorable prognosis in HCC. (**A**) Pan-cancer assessment of FBXO10 transcriptional levels across malignancies using the TIMER platform. Annotation: tumor and normal specimens are denoted in red and blue, respectively. Statistical significance determined with a two-tailed unpaired *t*-test. (**B**) UALCAN-based quantification reveals increased FBXO10 mRNA abundance in HCC clinical samples. Statistical significance determined with a two-tailed unpaired *t*-test. (**C**) A GEO dataset analysis confirms the upregulation of FBXO10 transcripts in HCC tissues. Statistical significance determined using Wilcox tests. (**D**) Comparative proteomic profiling in the HPA resource demonstrates enhanced FBXO10 expression in HCC versus adjacent normal liver tissues. (**E**) Survival probability curves from TCGA-LIHC (*n* = 365) illustrate significantly shorter overall survival in HCC patients with high FBXO10 expression. The survival analysis employed Kaplan–Meier curves with log-rank testing. * *p* < 0.05, ** *p* < 0.01, *** *p* < 0.001.

**Figure 2 cimb-47-00391-f002:**
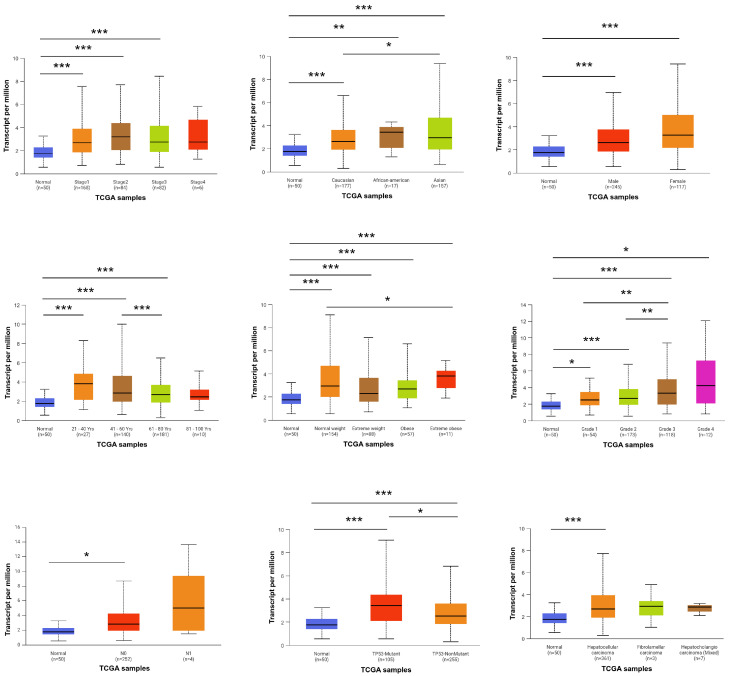
Relationship between FBXO10 expression level and clinicopathological characteristics of HCC patients. Statistical significance determined via one-way ANOVA with Bonferroni’s multiple comparisons test. * *p* < 0.05, ** *p* < 0.01, *** *p* < 0.001.

**Figure 3 cimb-47-00391-f003:**
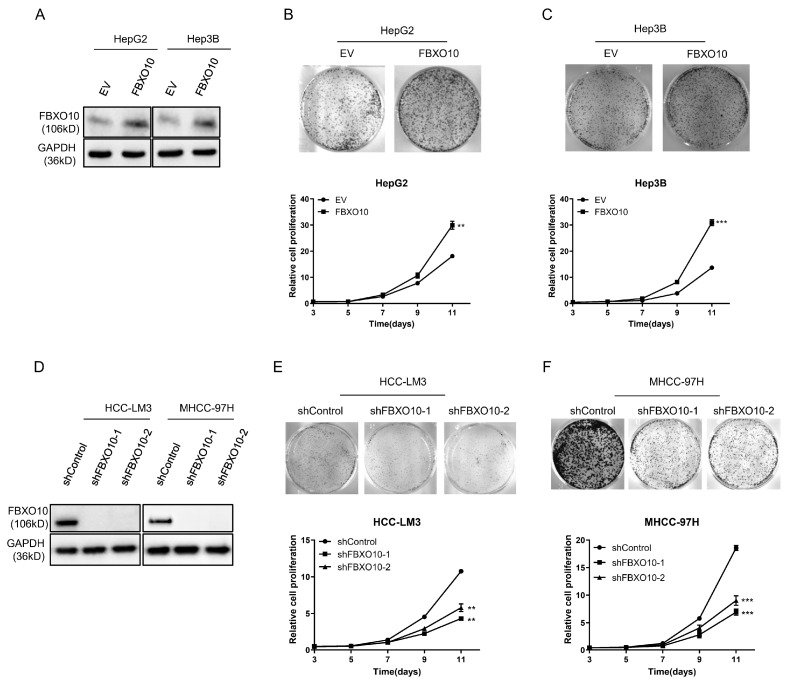
FBXO10 enhances hepatocellular carcinoma cell proliferation. (**A**) Western blot analysis demonstrating FBXO10 overexpression in HepG2 and Hep3B cells, with GAPDH serving as the loading control. (**B**,**C**) Morphological images and proliferation kinetics of HepG2 and Hep3B cells after ectopic FBXO10 expression. (**D**) Validation of FBXO10 knockdown efficiency via immunoblotting using two independent shRNAs in HCC-LM3 and MHCC-97H cells, normalized to GAPDH. (**E**,**F**) Morphological documentation and proliferation curves of HCC-LM3 and MHCC-97H cells following FBXO10 depletion. Data represent mean ± SD of triplicate experiments; statistical analysis performed using a two-tailed unpaired *t*-test. ** *p* < 0.01, *** *p* < 0.001.

**Figure 4 cimb-47-00391-f004:**
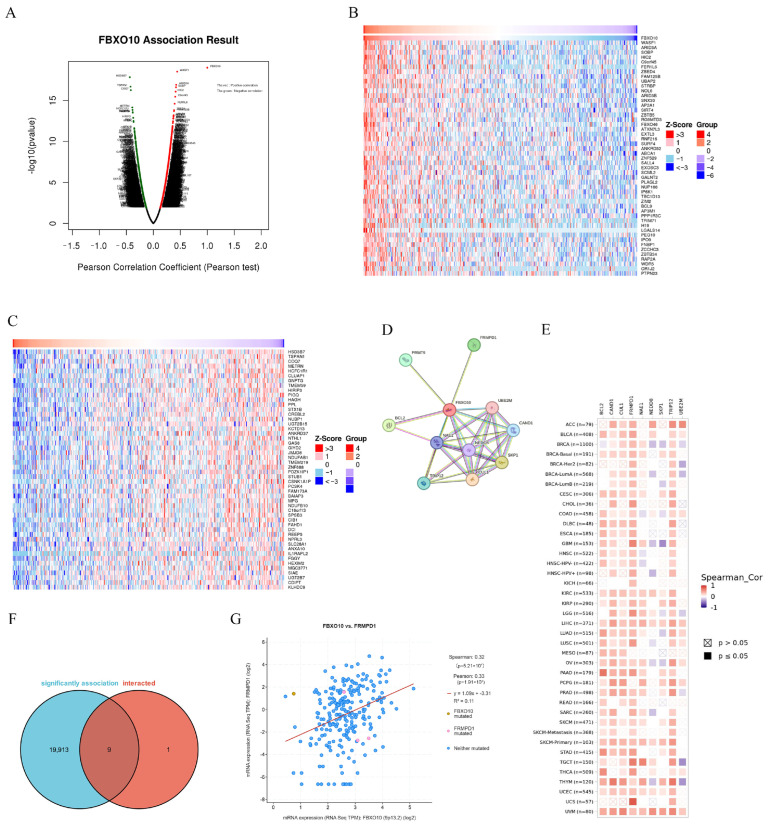
PPI network analysis of FBXO10-related genes. (**A**) The LinkedOmics database was utilized to identify genes exhibiting significant associations with FBXO10 in HCC specimens. (**B**,**C**) Heatmaps illustrate the top 50 genes demonstrating positive (**B**) and inverse (**C**) correlations with FBXO10 expression patterns in HCC. (**D**) PPI network mapping of FBXO10-associated genes, with molecular relationships visualized through STRING database interrogation. (**E**) Comparative analysis revealing overlapping genetic signatures between FBXO10-coexpressed genes and its direct protein interactors. (**F**) Expression covariation between FBXO10 and FRMPD1 quantitatively evaluated using cBioPortal’s analytical platform. (**G**) Pan-cancer correlation profiling of FBXO10 with multiple gene targets across malignancies conducted through the TIMER 2.0 computational framework. Pearson’s test was selected for linear correlations in normally distributed data, whereas Spearman’s test was applied to nonparametric or rank-based associations.

**Figure 5 cimb-47-00391-f005:**
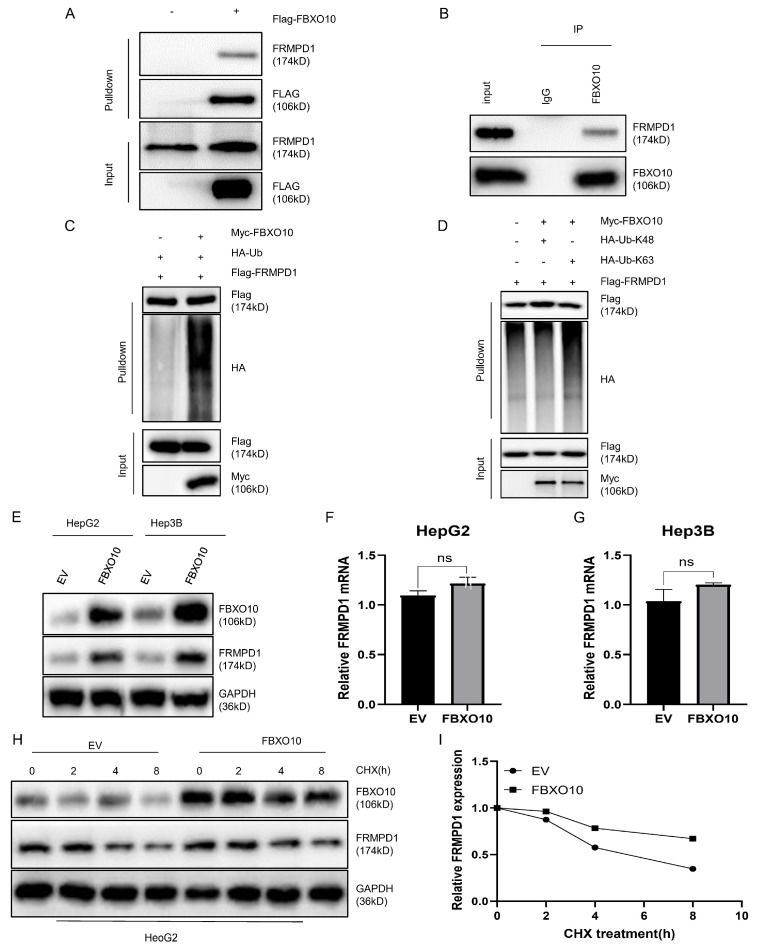
FBXO10 facilitates the K63-linked ubiquitination of FRMPD1 and increases its protein stability. (**A**) HEK-293T cells transiently transfected for 24 h with Flag-tagged FBXO10 plasmids underwent anti-Flag immunoprecipitation, followed by immunoblotting analysis with FRMPD1 and Flag antibodies. (**B**) Endogenous interaction between FBXO10 and FRMPD1 was confirmed in HepG2 cells through co-immunoprecipitation using FBXO10-specific antibodies. (**C**,**D**) Following 6 h pretreatment with the 10 μM proteasome inhibitor MG132, HEK-293T cells transfected with designated plasmids were analyzed via HA-affinity pulldown or direct immunoblotting with specified antibodies. (**E**) Western blot detection of FRMPD1 protein levels in FBXO10-overexpressing HepG2 and Hep3B cell lines. (**F**,**G**) Quantitative RT-PCR analysis of FRMPD1 mRNA expression in FBXO10-transfected HepG2 and Hep3B cells. (**H**) Cycloheximide (40 μM) chase assay monitoring time-dependent FRMPD1 degradation in FBXO10-expressing versus empty vector control HepG2 cells through immunoblotting. GAPDH served as the internal normalization control. (**I**) Quantitative densitometry of FRMPD1 protein levels relative to GAPDH (analyzed by ImageJ software Fiji 2.9.0). Results shown as mean ± SD from three independent experiments. Statistical comparisons were performed using two-tailed Student’s *t*-tests (ns: non-significant).

**Figure 6 cimb-47-00391-f006:**
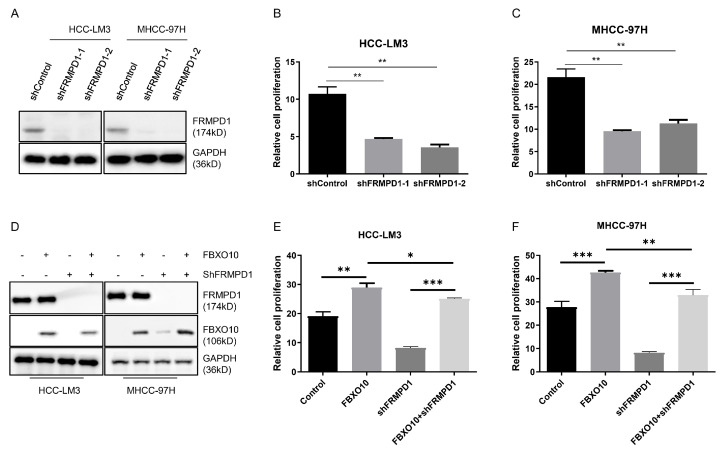
FBXO10 promotes hepatocellular carcinoma cell proliferation through FRMPD1 regulation. (**A**) Western blot confirmation of FRMPD1 knockdown efficacy in HCC-LM3 and MHCC-97H cells transfected with two independent FRMPD1-specific shRNAs. (**B**,**C**) Proliferation analysis using CCK-8 assay conducted 48 h after cell plating in 96-well plates following FRMPD1 suppression. (**D**) Simultaneous transfection of FBXO10-overexpressing plasmids and FRMPD1-specific shRNA constructs into HCC-LM3 and MHCC-97H cell lines. (**E**,**F**) Functional assessment of FRMPD1 knockdown on FBXO10-mediated proliferative effects through CCK-8 analysis. Experimental results from three independent replicates displayed as mean ± SD. Statistical significance determined via one-way ANOVA with Bonferroni’s multiple comparisons test. * *p* < 0.05, ** *p* < 0.01, *** *p* < 0.001.

## Data Availability

The original contributions presented in this study are included in this article. Further inquiries can be directed to the corresponding authors.

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
