# Peer review of "FBXO10 Drives Hepatocellular Carcinoma Proliferation via K63-Linked Ubiquitination and Stabilization of FRMPD1"

_cimb, 2025, doi:10.3390/cimb47060391_

Round 1

Reviewer 1 Report

Comments and Suggestions for Authors

The abstract is generally well-structured and follows a logical flow. However, some sentences could be refined for conciseness and clarity. For example, combining or simplifying longer sentences may improve readability.

1: The use of databases (TIMER, UALCAN, GEO) is noted, but it's not clear if these datasets were integrated or used independently. Clarifying this would strengthen the methods description.

2: The identification of the FBXO10–FRMPD1 axis as a novel mechanism is interesting. Consider emphasizing more clearly how this finding advances the current understanding of ubiquitination in HCC.

3: While the term “multi-omics” is used, the abstract mainly discusses transcriptomics. If other omics (e.g., proteomics, epigenomics) were used, briefly mention them for accuracy and completeness

4 : some literature reference 

Analysis and prediction of hematocrit in microvascular networks Temporal-spatial heterogeneity of hematocrit in microvascular networks In silico and in vitro study of the adhesion dynamics of erythrophagocytosis in sickle cell disease

Author Response

Dear Editor and Reviewers,

       Thank you for your letter and for the reviewers’ comments concerning our manuscript entitled “FBXO10 Drives Hepatocellular Carcinoma Proliferation via K63-Linked Ubiquitination and Stabilization of FRMPD1” (ID: cimb-3604126). Those comments are all valuable and very helpful for revising and improving our paper, as well as the important guiding significance to our researches. We have studied comments carefully and have made correction which we hope meet with approval. The revised manuscript incorporates both content edits (highlighted in red) and language refinements (highlighted in blue) via MDPI Author Services, with an editing certificate attached for verification. We hope that the new manuscript will meet your magazine’s standards. Below you will find our point-by-point responses to the reviewers’ comments/questions:

The abstract is generally well-structured and follows a logical flow. However, some sentences could be refined for conciseness and clarity. For example, combining or simplifying longer sentences may improve readability.

Response: Thank you for your constructive feedback on our manuscript. We have carefully addressed the reviewer's suggestion to improve conciseness and clarity in the abstract. The revised text has undergone professional English editing through MDPI Author Services (certificate attached) after getting your advice. All modifications are highlighted in blue in the revised manuscript. We believe the revised version now better aligns with journal standards. Thank you again for your valuable input. We remain available for any additional modifications required.

1: The use of databases (TIMER, UALCAN, GEO) is noted, but it's not clear if these datasets were integrated or used independently. Clarifying this would strengthen the methods description.

Response: We thank the reviewer for this important observation.

To clarify: TIMER provided a pan-cancer overview of FBXO10 dysregulation, guiding our focus to HCC; UALCAN independently validated and expanded TCGA-derived FBXO10 expression patterns in HCC, correlating them with clinicopathological parameters; GEO (GSE76427) served as an independent external cohort to confirm FBXO10 expression trends observed in TCGA. No raw dataset integration was performed; each database was used sequentially for hypothesis generation (TIMER), validation (UALCAN), and external confirmation (GEO). This tiered approach ensured analytical rigor and reduced platform-specific bias. We have revised the Methods section (2.1–2.3) to explicitly state this workflow. (Page 2-3 Line 77-95)

2: The identification of the FBXO10–FRMPD1 axis as a novel mechanism is interesting. Consider emphasizing more clearly how this finding advances the current understanding of ubiquitination in HCC.

Response: Thank you for your constructive feedback.

We have revised the Discussion section to explicitly emphasize how the FBXO10-FRMPD1 axis advances the understanding of ubiquitination in HCC. Key revisions include:

  1. Mechanistic Novelty: We highlight that FBXO10 stabilizes FRMPD1 via K63-linked ubiquitination, a non-canonical role diverging from the classical proteasomal degradation mediated by most F-box proteins (e.g., FBXL16).
  2. Context-Dependent Ubiquitination: We underscore the tissue-specific duality of FBXO10, which acts as a tumor suppressor in lymphoid malignancies via substrate degradation but drives HCC progression through FRMPD1 stabilization. This illustrates how microenvironmental cues reprogram ubiquitination pathways to serve opposing roles in distinct cancers.
  3. Therapeutic Implications: We propose that targeting the FBXO10-FRMPD1 interaction or K63 ubiquitination could offer novel strategies for HCC treatment, complementing current proteasome-focused therapies.

These revisions clarify how our findings redefine ubiquitination’s role in HCC pathogenesis and highlight its potential as a precision oncology target. (Page 13-14 Line 374-428)

3: While the term “multi-omics” is used, the abstract mainly discusses transcriptomics. If other omics (e.g., proteomics, epigenomics) were used, briefly mention them for accuracy and completeness

Response: We agree with this important technical distinction.

The terminology has been standardized to "bioinformatics analyses" throughout the manuscript, with explicit clarification in the Methods section regarding the study’s scope. (Page 1 Line 18)

4: some literature reference

Analysis and prediction of hematocrit in microvascular networks Temporal-spatial heterogeneity of hematocrit in microvascular networks In silico and in vitro study of the adhesion dynamics of erythrophagocytosis in sickle cell disease

Response: Thank you for your attention to the relevant literature.

We confirm that we have appropriately cited the following studies in our revised manuscript:

  1. "Analysis and prediction of hematocrit in microvascular networks" (referenced as [38]) to contextualize hemodynamic heterogeneity in tumor microenvironments.
  2. "In silico and in vitro study of the adhesion dynamics of erythrophagocytosis in sickle cell disease" (referenced as [39]) to highlight biophysical parallels in cell-cell interactions.

These citations strengthen our discussion of microenvironmental influences on cancer progression. We appreciate your thorough review and have ensured all references are correctly formatted and contextualized. (Page 13 Line 394-398)

Reviewer 2 Report

Comments and Suggestions for Authors

I had the opportunity to review the manuscript entitled FBXO10 Drives Hepatocellular Carcinoma Proliferation via K63-Linked Ubiquitination and Stabilization of FRMPD1 by Liu et al. The authors tried to use bioinformatics and molecular biology methods in their research. However, I have some comments and a lot of questions for the biological part as follows.

  1. Were cells cultured without antibiotics? (Section 2.8)
  2. Where are the HEK-293T and MG132 cell lines? How were they cultured? I ask because the authors used those lines in their study (Figure 5).
  3. Add the dilution of antibodies used in the study. In addition, fill in the catalogue number of secondary antibodies and their dilutions.
  4. How were the knockdown cells obtained? Was lipofection/electroporation used? Perhaps another method. It wasn’t mentioned.
  5. How were the cells fixed (Section 2.12). What microplate reader was used to measure the absorbance (Section 2.12)?  
  6. What is the composition of SDS buffer (Section 2.14)? Was it Laemmli buffer?
  7. I wonder what was ‘normal’ in Figure 2? For example, when you examine the relationship between FBXO10 level and sex, were ‘normal’ results obtained from male or female? What with other results?
  8. In the MHCC-97H cell there is no FBXO10 silencing. The level of FBXO10 protein in shFBXO10-1 and shFBXO10-2 is only slightly lower. Further results are false positive. The same situation applies to the case of MHCC-97H and FRMPD-1 knockdown.
  9. In case of HCC-LM3 the authors should only use shFBXO10-2. The shFBXO10-1 cells were also not silenced. I recommend deleting these results.
  10. Did the authors analyse FRMPD1 overexpression only? If not, it MUST be analysed. Perhaps overexpression of FRMPD1 will also increase cell proliferation, and the effect you observed is mostly caused by this action.
  11. The authors must also conduct a literature study. An extended discussion is needed.
  12. I also suggest adding limitations of the study and future perspectives.
  13. I would like to see all the whole membranes with marker. The authors mentioned that the experiments were performed in triplicates. How do the authors know if they are analysing the right product if they don't have a size marker?

Round 2

Reviewer 2 Report

Comments and Suggestions for Authors

The authors responded to most of my comments and the quality of the manuscript significantly increase. However, I would like to see ALL Western blot membranes from ALL the experiments in triplicated. You mentioned that the experiments have been performed in triplicated. It means that you should have 3 membranes for FBXO10 protein in line HepG2, 3 for GAPDH for this cell line, 3 for Hep3B for analysed protein and 3 for GAPDH and so on. Not only 2 that you attached to the responses.
